# Blunt Trauma in Children: Efficacy and Safety of Transarterial Embolization, 10-Year Experiences in a Single Trauma Center

**DOI:** 10.3390/diagnostics13213392

**Published:** 2023-11-06

**Authors:** Seyoung Ko, Hoon Kwon, Chang Won Kim, Hojun Lee, Jae Hun Kim, Hohyun Kim, Chan Ik Park

**Affiliations:** 1Department of Radiology, Pusan National University Hospital, Pusan National University School of Medicine, Pusan National University, Busan 49241, Republic of Korea; 2Department of Trauma Surgery and Surgical Critical Care, Pusan National University Hospital, Pusan National University School of Medicine, Pusan National University, Busan 43241, Republic of Korea

**Keywords:** embolotherapy, pediatrics, trauma

## Abstract

Background: Transcatheter arterial embolization (TAE) is an established approach for controlling hemorrhage in adults with acute abdominal and pelvic trauma. However, its application in pediatric trauma is not well established. This study aimed to evaluate the safety and effectiveness of TAE in a population of pediatric patients with blunt trauma. Methods: This retrospective study was conducted in pediatric patients (<18 years) who underwent TAE for blunt trauma between February 2014 and July 2022. The patients were categorized into subgroups based on age and body weight. Patient demographics, injury severity, transfusion requirements, and clinical outcomes were analyzed. Results: Exactly 73 patients underwent TAE. Technical success was achieved in all patients (100%), and clinical success was achieved in 83.6%. The mortality and complication rates were 4.1% and 1.4%, respectively. The mean duration of hospitalization was 19.3 days. Subgroup analysis showed that age, body weight, and sex did not significantly affect clinical success. The injury severity score and transfusion requirement were predictors of clinical success, with lower values associated with better outcomes. Conclusions: TAE is effective and safe for managing blunt pediatric trauma in younger and lighter patients. Injury severity and transfusion requirement are predictors of clinical success.

## 1. Introduction

In adults, transcatheter arterial embolization (TAE) is an established minimally invasive approach to achieve rapid hemorrhage control in acute blunt or penetrating abdominal and pelvic trauma [1,2,3,4]. However, the indications for TAE in children with trauma are still not well established, and the major mechanism of trauma in children is blunt injury. Although some studies have reported successful TAE in pediatric patients with blunt trauma, the number of studies and associated sample sizes are few [5,6,7]. Furthermore, in previous studies, the categorization of individuals aged 18 and below was limited to the term “pediatric patients.” However, considering the rapid growth during adolescence, various distinct physical characteristics have emerged within the pediatric population itself. These nuances must be adequately taken into consideration in the management of pediatric patients. Moreover, the previous studies have exclusively focused on Western pediatric patients. In our research, we aimed to investigate potential differences in transcatheter arterial embolization for pediatric trauma, specifically in adolescents and younger children, within the context of Eastern populations.

Thus, the focus of this study was to assess the safety and effectiveness of transcatheter arterial embolization (TAE) in a substantial cohort of pediatric patients with blunt trauma.

## 2. Materials and Methods

### 2.1. Participants

This retrospective single-center study enrolled pediatric patients with blunt injury between February 2014 and July 2022 at Pusan National University Hospital.

The National Institutes of Health (NIH) defines pediatric patients as those below 18 years. Therefore, patients below 18 years who underwent TAE due to arterial injury in the chest, abdominal, and pelvis, as observed on computed tomography (CT) images, were enrolled in this study. We obtained patients’ electronic medical records and radiologic images (CT, angiography, and embolization).

All patients in this study (referred to as ‘pediatrics’) were classified into two groups according to their age, following the NIH categorization:

(1) Patients below 12 years (‘Child’ group).

(2) Patients above 12 years (‘Adolescent’ group).

Patients were also classified into two groups based on the median value of their body weight, namely ‘heavy’ and ‘light’ groups.

### 2.2. Patient Data

All patient data from the Korea Trauma Bank and electronic medical records were reviewed, including age, sex, body weight, initial vital signs, Injury Severity Score (ISS), transfusion requirement before embolization, overall hospitalization, and embolization complications.

### 2.3. CT Images

All patients included in this study were initially evaluated by a trauma surgery team based on the CT images. The inclusion criteria for TAE were as follows:

(1) Solid organ injury with contrast blush on CT.

(2) Pelvic injury with contrast blush on CT.

(3) Solid organ or pelvic bone injury without contrast blush, with suspected ongoing hemorrhage or large hematoma on CT.

### 2.4. Angiography and Embolization

Angiography and embolization were performed in the interventional radiology suite at the trauma center. Arterial access was achieved via the common femoral artery under fluoroscopic and ultrasonographic guidance. A 21-gauge micropuncture needle set (Cook, Bloomington, Indiana) was used for the initial arterial puncture, and a 4-F for 5-F sheath was used to maintain arterial access throughout the procedure [1]. A 4-F or 5-F Cobra catheter and a coaxial 1.7-F or 1.9-F micro-catheter were used for angiography and embolization. On angiography, we classified arterial injuries into four types:

(1) Arterial transection: A truncated artery.

(2) Extravasation: Leakage of intravascular contrast media leak from the vessel to the surrounding soft tissue.

(3) Pseudo-aneurysm: A bleeding focus forming a sac-like shape.

(4) Petechial hemorrhage: A small dot-like hemorrhage.

The injured vessels and segmental branches were superselected and embolized using a gel foam slurry, metal coils, or n-butyl-2-cyanoacrylate (NBCA). Post-embolization angiography was always obtained.

### 2.5. Embolization Outcomes

We evaluated the effectiveness of the embolization procedure using two criteria: (1) technical success, defined as the complete disappearance of bleeding on post-embolization angiography [8]; (2) clinical success, which involves hemodynamic stabilization without additional surgery or second-session angiography for bleeding control [9].

### 2.6. Complications

The safety of the embolization procedure was evaluated based on the incidence rate of embolization-related complications. Complications were classified as major or minor, according to the Society of Interventional Radiology Standards of Practice Committee guidelines. Major complications were defined as those requiring major therapy, necessitating an unplanned increase in level of care or prolonged hospitalization (>48 h), or resulting in permanent adverse sequelae or death. Minor complications were defined as those requiring no or minimal therapy, including overnight admission for observation only [10,11].

### 2.7. Statistical Analysis

To identify the predictors of clinical success or failure, the patients were divided into two groups based on the definition of clinical success. Additionally, patients’ demographics (age, ISS, amount of transfusion before embolization, body weight, and sex) were compared between the groups. Independent *t*-test, Pearson’s chi-square test, and logistic regression analysis were used for statistical analyses using SPSS 27.0.

## 3. Results

In total, 74 patients below 18 years underwent angiography for blunt trauma between February 2014 and July 2022. One patient who did not undergo embolization showed a liver laceration on CT, but there was no evidence of active bleeding on hepatic artery angiography. Thus, we focused on 73 patients who underwent embolization. The demographics of all patients who underwent arterial embolization are presented in Table 1.

The 73 patients who underwent angiography for chest, abdomen, and pelvis trauma required embolization involving 103 sites. Table 2 summarizes the embolization performed in total patients.

Technical success was achieved in all patients (100%), and clinical success was achieved in 62 of the 73 patients (83.6%). Clinical success was not achieved in 11 patients: 6 underwent secondary angiography and embolization, while 5 underwent additional surgery after embolization to control bleeding. The mortality rate was 4.1% (n = 3); two patients died after additional surgery, and one died after embolization. The complication rate was 1.4% (n = 1), and one patient underwent an additional left hemi-hepatectomy due to ischemic hepatitis after embolization. The mean duration of hospitalization in all patients was 19.3 days (median duration 11, 1–118). Table 3 summarizes the clinical outcomes of all patients.

Table 4 presents a comparison of patient demographics among the four groups classified based on age and body weight.

Table 5 shows a comparison of TAE clinical outcomes among the four groups.

In the ‘Child’ group, technical success was achieved in all patients (100%), and clinical success was achieved in 27 of the 29 patients (93.1%). Clinical success was not achieved in two patients: one underwent secondary angiography and embolization, while the other underwent additional surgery after embolization but died after surgery.

In the ‘Adolescent’ group, technical success was achieved in all patients (100%), and clinical success was achieved in 34 of the 44 patients (77.3%). Clinical success was not achieved in 10 patients: 55 underwent secondary angiography and embolization, while 4 underwent additional surgery after embolization to control bleeding. The mortality rate was 4.5% (n = 2); one patient died after additional surgery (exploratory thoracotomy), while the second patient died after embolization. The complication rate was 2.3% (n = 1), and one patient underwent an additional left hemi-hepatectomy due to ischemic hepatitis after embolization.

In the ‘Light’ group, technical success was achieved in all patients (100%), and clinical success was achieved in 33 of the 39 patients (84.6%). Of the remaining six who did not achieve clinical success, two patients underwent secondary embolization, another underwent secondary angiography, and three underwent additional surgery after embolization to control bleeding. The mortality rate was 7.7% (n = 3); one patient died after embolization, while the other two patients died after post-embolization surgery. The complication rate was 0% (n = 0).

In the ‘Heavy’ group, technical success was achieved in all patients (100%), and clinical success was achieved in 28 of the 34 patients (82.4%). Three patients underwent secondary angiography and embolization. The other three patients underwent additional surgery after embolization for bleeding control. The mortality rate was 0%. The complication rate was 2.9% (n = 1), and the patient underwent a left hemihepatectomy due to ischemic hepatitis.

There were no significant differences in clinical success among the four groups (child, adolescent, light, and heavy) (*p* = 0.238).

Table 6 and Table 7 summarize the predictors of clinical outcomes. Patient demographics were assessed for univariate associations with effective hemorrhage control. The ISS and amount of transfusion before embolization could be predictors of clinical outcome; they were significantly lower in the clinical success group (*p* = 0.006 and *p* < 0.001, respectively). There were no significant differences in age, body weight, and sex between the clinical and nonclinical success groups. (*p*-value: 0.472, 0.691, 0.224, respectively) (Table 6).

After conducting logistic regression analysis, the significant differences observed between the clinical success group and the failure group in terms of ISS (injury severity score) and the amount of transfusion before angiography were further investigated, and the results have been summarized in Table 7. Interestingly, the analysis revealed that only the “amount of transfusion before angiography” variable showed statistical significance (*p* = 0.004) as a predictor of clinical outcome. The odds ratio was found to be 1.573, indicating that it had a significant impact on the likelihood of clinical success.

## 4. Discussion

This study evaluated the safety and efficacy of TAE in a large population of pediatric patients with blunt trauma. Technical success was achieved in all patients (100%), and clinical success was achieved in 83.6%. The mortality and complication rates were 4.1% and 1.4%, respectively. These findings underline the effectiveness and safety of TAE in managing blunt pediatric trauma across all age groups, including younger and lighter patients.

ISS and transfusion requirements exhibited significant differences between the clinical success and failure groups, with lower values indicating a more favorable treatment outcome. Particularly, among these factors, transfusion requirements were statistically significant and had a substantial impact on clinical outcomes. This observation aligns with experiences in adult trauma cases and various hemorrhagic conditions, as higher transfusion needs correspond to a greater extent of rapid and substantial bleeding. Furthermore, it is noteworthy to mention that patients with elevated ISS levels are more likely to exhibit increased transfusion demands. This correlation accentuates the heightened probability of patients falling into trauma-related coagulopathy, ultimately exacerbating prognostic implications. In cases of ongoing intra-abdominal bleeding post-trauma, it is recognized that mortality rates increase by 1% every 3 min [12]. This underlines the critical significance of swift hemostasis as a means to reduce the need for blood transfusions. Furthermore, in pediatric patients, as in adults, the approach of transarterial embolization gains even greater importance due to its capability of effectively addressing multiple sources of bleeding. With the rise of trauma interventional radiology’s efficacy in adults, the concept of damage control interventional radiology has emerged [13]. This approach prioritizes life-saving measures, leading to the swift mitigation of bleeding by minimizing procedural time and considering proactive, wide-ranging embolization. This concept finds particular relevance in the management of pediatric patients, where the potential for recovery holds even greater significance than in adults. Compared to adults, the advantages of intervention hold greater promise in addressing the multiple traumas seen in pediatric patients.

Trauma is a leading cause of death in children [1,6,14,15]. Despite this statistic, fewer resources and less attention have been directed toward treating injured children than toward injured adults [16]. This includes blunt thoracic and abdominal trauma, leading to hemodynamic instability associated with hemorrhage.

In adult trauma, the ability to treat life-threatening hemorrhages with TAE has spared countless patients from surgical morbidity [17]. However, in the current pediatric trauma treatment algorithm, angiography assessment with embolization plays a limited role [1,18]. Most children with solid organ injuries are managed with observation, and embolization is rarely performed, with limited indications [19]. Nevertheless, angiography and embolization are important in pediatric trauma due to the specific characteristics of these patients. At initial presentation, most pediatric patients do not display signs of profound shock, such as hypotension, tachycardia, or mental status changes [20,21]. Given this unique clinical attribute, a judicious approach becomes imperative when considering patient selection for angiography, advocating for a more proactive consideration of angiography and embolization. Additionally, imaging studies such as CT and focused assessment with sonography in trauma play a significant role in evaluating patients with trauma [22]. These imaging strategies not only aid in identifying potential injuries but also contribute to informed decision making regarding the necessity and appropriateness of interventions like angiography, enabling clinicians to tailor treatments to the specific needs of each pediatric trauma patient.

Previous studies have mainly dealt with solid and visceral organ injuries, and most studies include case series or small patient groups [5,7]. Lin et al. concluded that TAE is an alternative therapeutic modality for blunt renal injury in children with contrast medium extravasation on angiography in the kidney in 18 pediatric patients (six underwent TAE) [21]. Vo et al. concluded that angiography and embolization are relatively safe and potentially effective in a notably extensive population of 97 pediatric patients with abdominal and pelvic trauma (54 patients underwent TAE), marking it as one of the largest studies conducted in this area [1].

However, it is important to acknowledge that the existing studies in this field are not without their limitations. One notable limitation is the relatively small size of the patient groups that have been included in these studies. Furthermore, a crucial point of consideration is that the term ‘pediatrics’ has often been used to encompass individuals under the age of 18 without adequately accounting for the wide variability in physical characteristics among children. Given the rapid and diverse nature of growth during adolescence, it becomes imperative to recognize that children exhibit a range of physical traits. While some may resemble adults in certain physical aspects, others may not share the same similarities.

Another noteworthy limitation is the geographical and cultural scope of the previous research, which predominantly focused on Western populations. It is worth noting that Western adolescents tend to exhibit greater physical development in terms of height and body weight compared to their Eastern counterparts. This discrepancy raises an important question regarding the applicability of the findings from these studies to Eastern children. Simply extrapolating the safety conclusions drawn from Western studies to Eastern contexts might not be justified, as there could be intrinsic physiological differences between these populations that impact the safety and effectiveness of the therapeutic interventions under consideration.

In essence, while previous research has provided valuable insights into the use of therapeutic interventions in pediatric trauma cases, these findings must be interpreted within the context of their limitations. In order to establish a more comprehensive and universally applicable understanding, it becomes crucial to conduct studies that encompass a wider range of patient demographics, accounting for the diverse physical characteristics and developmental trajectories of children, while also encompassing different geographical and cultural backgrounds to ensure a more representative and informed perspective.

Compared to Vo’s report [1], this study included a relatively large number of patients (73 patients requiring embolization) and a significantly lower median age (16 years vs. 14 years), body weight (68 kg vs. 50 kg), and mortality (12% vs. 3%). Additionally, the patients had a mean ISS of 22.5, and almost all patients had major trauma. Our findings demonstrate no significant difference in clinical success in ‘child,’ adolescent,’ light,’ and ‘heavy’ groups. This suggests that TAE is an effective treatment even in ‘real children’ who are much younger and with lower body weight.

In adult chest trauma, intercostal and internal mammary artery injuries may cause hemothorax and mediastinal hemorrhage, which can be treated with embolization [20,23,24]. However, few studies have reported on the usefulness of embolization for treating chest trauma in adults. This study also included five cases of hemothorax and hemomediastinum that were effectively treated with embolization in pediatric patients with chest trauma. Each patient was effectively treated with embolization of the internal mammary, right bronchial, and intercostal arteries. Additionally, one patient with a gastroduodenal artery pseudo-aneurysm associated with blunt abdominal trauma was successfully treated with coil embolization. These cases show that treatment with TAE is not limited to solid organ or pelvic injury but can be applied as a minimally invasive method to treat various hemorrhages and vascular injuries.

Radiation exposure is an important concern in the pediatric population [25,26,27], as children have a significantly higher risk of radiation-induced malignancies than adults [28,29]. However, the authors endeavored to minimize radiation exposure according to ALARA (“as low as reasonably achievable”) concepts when performing TAE [30,31,32]. We also believe that the benefit from TAE outweighs the risk of radiation exposure.

Our study presents certain limitations. It was a retrospective study conducted at a single center, focusing solely on patients who underwent embolization, which could introduce a selection bias. Additionally, the absence of a comparison group comprising patients who received alternative surgical treatments or conservative management prevents a direct comparison of the therapeutic effects of embolization. However, this aspect is also influenced by the inherent nature of trauma patients, where the urgency of intervention and variability in the organs affected by trauma differ among the population. Moreover, trauma patients, unlike those with other conditions, possess diverse characteristics that make conducting prospective randomized control studies challenging from both clinical and ethical standpoints.

## 5. Conclusions

In summary, our study demonstrates that TAE is safe and effective in all pediatric patients with blunt trauma, regardless of age and body weight.

## Figures and Tables

**Table 1 diagnostics-13-03392-t001:** Patient demographics.

73 Patients		
Sex (Boys: Girls)		53:20
Age		
	Mean	13.1
	Median (range)	14 (2–18)
Body weight (kg)		
	Mean	48.8
	Median (range)	50 (13–96)
ISS		
	Mean	22.5
	Median (range)	21 (1–50)
Transfusion before angiography		43 (58.9)

ISS, injury severity score.

**Table 2 diagnostics-13-03392-t002:** Summary of trans-arterial embolization procedure in all patients.

73 Patients		
Target of embolization (103 sites in 73 patients)		
	Liver	20
	Spleen	35
	Kidney	17
	Pelvis	14
	Pancreas	1
	Thorax	6
	Lumbar spine and back muscle	4
	Others (sacrum, labia majora, and adrenal gland)	6
Angiographic findings		
	Extravasation	24
	Pseudo-aneurysm	13
	Arterial transection	6
	Petechial hemorrhage	19
	Others (Combined findings, arterior-portal shunt, and hyperemia)	11
Embolic agents		
	Gelatin sponge slurry	31
	Coils only	7
	NBCA	14
	Coils/Gelatin sponge slurry	12
	NBCA/Gelatin sponge slurry	4
	NBCA/Coil	1
	Gelatin sponge slurry/Coils/NBCA	3
	Autologous blood clot	1

NBCA, n-butyl-2-cyanoacrylate.

**Table 3 diagnostics-13-03392-t003:** Summary of clinical outcomes in all patients.

73 Patients	
Outcomes	Frequency (%)
Technical success	73 (100)
Clinical success	61 (83.6)
Death	3 (4.1)
Embolization-related complications	1 (1.4)
Hospitalization (days, mean)	19.3
Hospitalization (days, median(range))	11 (1–118)

**Table 4 diagnostics-13-03392-t004:** Comparison of patient demographics according to age and body weight.

	‘Child’ Group (n = 29)	‘Adolescent’ Group (n = 44)	‘Light’ Group (n = 39)	‘Heavy’ Group (n = 34)
Sex (Boys: Girls)	21:8	32:12	23:16	30:4
Age				
Mean	8.1	16.3	10.3	16.2
Median (range)	8.5 (2–12)	16.5 (13–18)	10 (2–18)	17 (12–18)
Weight (kg)				
Mean	31	60.2	33.5	66.4
Median (range)	26 (13–67)	57.5 (37–96)	37 (13–50)	64.5 (51–96)
ISS				
Mean	19.8	24.3	21.7	23.3
Median (range)	17 (4–45)	22 (1–50)	19 (1–45)	21.5 (9–5)
Transfusion before angiography (%)	14 (48.3)	29 (65.9)	24 (61.5)	19 (55.9)

**Table 5 diagnostics-13-03392-t005:** Comparison of clinical outcomes according to age and body weight.

	‘Child’ Group (n = 29)	‘Adolescent’ Group (n = 44)	‘Light’ Group (n = 39)	‘Heavy’ Group (n = 34)
Technical success (%)	29 (100)	44 (100)	39 (100)	34 (100)
Clinical success (%)	27 (93.1)	34 (77.3)	33 (84.6)	28 (82.4)
Death (%)	1 (3.4)	2 (4.5)	3 (7.7)	0 (0)
Embolization-related complications (%)	0 (0)	1 (2.3)	0 (0)	1 (2.9)
Hospitalization (days):				
Mean	11.5	24.5	16.3	15.6
Median	10 (1–33)	14.5 (2–118)	11 (1–118)	12 (2–91)

**Table 6 diagnostics-13-03392-t006:** Predictors of clinical outcome.

Predictors	Success (61)	Failure (12)	*p*-Value
Age	12.7	14.7	0.472
ISS	20.8	30.8	0.006
Transfusion before angiography	1.1	5.8	<0.001
Body weight (kg)	48.5	50.3	0.691
Sex	46:15	7:5	0.225

ISS: injury severity score.

**Table 7 diagnostics-13-03392-t007:** Logistic regression analysis of predictors of clinical outcomes.

Predictors	B	*p*-Value	Exp(B)
ISS		0.388	
Transfusion before angiography	0.453	0.004	1.573

## Data Availability

The data are available in this article.

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
