# Peer review of "Blunt Trauma in Children: Efficacy and Safety of Transarterial Embolization, 10-Year Experiences in a Single Trauma Center"

_diagnostics, 2023, doi:10.3390/diagnostics13213392_

Round 1

Reviewer 1 Report

Comments and Suggestions for Authors

The main question addressed by the research is the efficacy and treatment of transarterial embolization for blunt trauma in children.

The topic is original and interesting because little is known about this topic. Moreover, it's a 10-year experience that means it is a long study period.

Since there are only a few studies on this topic, it adds value to the TAE for children.

I think they should add a multivariate analysis to better understand which factors are predictive of success and to confirm their data.

References are appropriate.

Comments on the Quality of English Language

No comments

Author Response

We appreciate the valuable feedback provided by the reviewer regarding our research on transarterial embolization (TAE) for blunt trauma in children.

In response to the suggestion to include a multivariate analysis, we have incorporated a binary logistic regression analysis into our study.

This addition will allow us to identify predictive factors better and further validate our findings. Thank you for your insightful recommendations, which have strengthened the depth and reliability of our research.

The modified content is highlighted in yellow in the annotated copy.

Reviewer 2 Report

Comments and Suggestions for Authors

It is a very interesting paper. Nevertheless, some points have to be clarified:

- clarify the definition of light and heavy groups in materials and methods

- explain the statistic that you used to find predictive factors in the methods section (there is only something in the results)

The study is interesting but I think the statistics should be improved with a multivariate analysis to confirm the results

Author Response

Comments 1:  

- clarify the definition of light and heavy groups in materials and methods

- explain the statistic that you used to find predictive factors in the methods section (there is only something in the results)

The study is interesting but I think the statistics should be improved with a multivariate analysis to confirm the results

Response 1:

We appreciate the valuable feedback provided by the reviewer regarding our research on transarterial embolization (TAE) for blunt trauma in children.

As per your feedback, we have provided further clarification regarding the heavy and light groups.

Comments 2: -explain the statistic that you used to find predictive factors in the methods section (there is only something in the results)

Response 2:

Thank you for the valuable suggestion.

We have addressed the comment regarding the insufficient statistical explanation in the manuscript by adding further details and clarification.

The modified content is highlighted in yellow in the annotated copy.

Comments 3: The study is interesting but I think the statistics should be improved with a multivariate analysis to confirm the results

Response 3:

Thank you for the valuable suggestion.

In response to the comment suggesting the inclusion of multivariate analysis, we have conducted additional logistic regression analysis.

This addition will allow us to better identify predictive factors and further validate our findings. Thank you for your insightful recommendations, which have strengthened the depth and reliability of our research.

Round 2

Reviewer 2 Report

Comments and Suggestions for Authors

I think the Authors have clarified all the issues